# Guidongnins I–J: Two New 6,7-*seco*-7,20-Olide-*ent*-kaurene Diterpenes with Unusual Structures from *Isodon rubescens*

**DOI:** 10.3390/ijms241713451

**Published:** 2023-08-30

**Authors:** Juan Zou, Jianghai Ye, Chenliang Zhao, Jingjie Zhang, Yahua Liu, Lutai Pan, Kang He, Hongjie Zhang

**Affiliations:** 1School of Pharmacy, Guizhou University of Traditional Chinese Medicine, 4 Dongqing Road, Guiyang 550025, China; zoujuan466@gzy.edu.cn (J.Z.); yejianghai013@gzy.edu.cn (J.Y.); zhaochenliang014@gzy.edu.cn (C.Z.); zhangjingjie047@gzy.edu.cn (J.Z.); liuyahua145@gzy.edu.cn (Y.L.); panlutai489@gzy.edu.cn (L.P.); 2School of Chinese Medicine, Hong Kong Baptist University, 7 Baptist University Road, Hong Kong SAR 999077, China

**Keywords:** *Isodon rubescens*, *ent*-kaurene diterpenoid, isopropyl group, anti-cancer, plausible biosynthetic pathway

## Abstract

Two undescribed *ent*-kaurene diterpenes, named guidongnins I (**1**) and J (**2**), were isolated from the medicinal plant *Isodon rubescens*. Compound **1** was determined to contain an unprecedented 23 carbons in the skeleton by bearing an extra isopropyl group at C-17 out of the diterpenoid parent structure, and compound **2** was the first example of 6,7-*seco*-7,20-olide-*ent*-kaurenes with two fused-tetrahydrofuran rings formed between C-6 and C-19/C-20 through oxygen bridges. Their structures, including their absolute configurations, were determined using the analyses of the spectroscopic and X-ray diffraction data. Guidongnins I (**1**) and J (**2**) were assessed for their anti-cancer activities against the growth of various cancer cell lines, and **2** displayed cytotoxic potency against HepG2 at IC_50_ 27.14 ± 3.43 μM.

## 1. Introduction

The *Isodon* genus, belonging to the Labiatae family, is renowned for its abundant diterpenes featuring diverse carbon frameworks, which exhibit a wide array of biological activities. Over 1200 new diterpenes have been identified from the genus previously [1,2,3,4,5]. Approximately 30 species in this genus were constantly used as Chinese folk medicines. Among them, *I. rubescens* was found to possess multiple pharmacological effects that have been used to treat respiratory problems, gastrointestinal inflammation, bacterial infected diseases, and cancer [5], and its refined extract has been developed as a Chinese patent medicine to treat sore throat, bacterial infection, and cancer [6]. Many interesting diterpenes have been isolated from this plant [5,7,8,9,10,11]. Apart from the common *ent*-kaurane diterpenoids, 6,7-*seco*-*ent*-kaurane diterpenoids are constantly found in this species and they have been demonstrated to show significant antiproliferative activities against various human tumor cell lines. Acetylexidonin [12] and lushanrubescensin H [13] are the two examples that displayed potent antitumor activities. To further explore the novel bioactive diterpenoids bearing this skeleton from *I. rubescens*, our group has systematically investigated the phytochemicals of the plant species collected from Guizhou. A number of novel diterpenoids, such as rubesanolides A-G, have been identified from this plant in our previous studies [7,8,9], which revealed that *I. rubescens* is a special *Isodon* plant that could richly produce diterpene metabolites with unique structures. As part of our ongoing efforts to find new bioactive diterpenoids, we isolated guidongnins I (**1**) and J (**2**) as two novel *ent*-kaurane diterpenoids from this species (Figure 1). Compound **1** was found to be the first example of a novel C23-carbon skeleton bearing a 6,7-*seco* diterpene framework conjugated with an isopropyl group at C-17, and **2** contained a complex ring system with two unprecedented tetrahydrofurans fused with a *δ*-lactone ring. The two compounds could be derived from a commonly occurring *ent*-kaurane diterpene via a series of enzyme-catalyzed biosynthetic conversions outlined in Figure 1. This new discovery further illuminates the chemical diversity of *ent*-kaurene diterpenoids in this species. By evaluating their bioactivities against cancer cells A-549, HL-60, MCF-7, HepG2, and COLO-205, **2** was found to be specifically active against the HepG2 (liver) with an IC_50_ value of 27.14 ± 3.43 μM. Therefore, we report the isolation, structure elucidation, biosynthetic pathways, along with the bioactivities of two new diterpenoids (**1**–**2**).

## 2. Results and Discussion

### 2.1. Compound Identification

Dried leaves of *I. rubescens* were subjected to MeOH extraction, yielding a corresponding MeOH extract, which was chromatographed with RP-8 and silica gel chromatography to yield guidongnin I (**1**) and guidongnin J (**2**).

Compound **1** was isolated and crystalized as block crystals in MeOH (m.p. 247–248 °C), ${\left[\alpha \right]}_{D}^{20}$ −127.27 (*c* 0.22, MeOH). It was identified to have the molecular formula of C_23_H_32_O_6_ elucidated from its HRESIMS analyses (found *m*/*z* 427.2091 [M + Na]^+^, calcd. for C_23_H_32_O_6_Na, 427.2086), indicating eight degrees of unsaturation. The IR absorption of **1** displayed the presence of hydroxyl (*ν*_max_ 3380 cm^−1^) and carbonyl (1697, 1749 cm^−1^) groups. The ^13^C NMR (Table 1) data of **1** displayed 23 carbon signals, which were identified to belong to three non-oxy quaternary carbons [*δ*_C_ 41.2 (C-4), 56.1 (C-8), 39.2 (C-10)], a ketone group [*δ*_C_ 215.9 (C-15)], two lactones [*δ*_C_ 177.0 (C-6), 174.9 (C-7)], five non-oxy methines [*δ*_C_ 47.7 (C-5), 43.3 (C-9), 32.0 (C-13), 52.8 (C-16), 27.3 (C-21)], an oxy methine [*δ*_C_ 65.2 (C-11)], six non-oxy methylenes [*δ*_C_ 28.5 (C-1), 19.3 (C-2), 33.6 (C-3), 32.3 (C-12), 36.3 (C-14), 37.3 (C-17)], two oxy methylenes [*δ*_C_ 76.9 (C-19), 72.9 (C-20)], and three methyl groups [*δ*_C_ 22.1 (C-18), 21.8 (C-22), 23.7 (C-23)]. On the basis of the literature data [14] and the chemotaxonomic considerations, **1** was determined to have a 6,7-*seco*-7,20-olide-*ent*-kaurane skeleton structure, which was confirmed by the analysis of the 2D NMR correlation data (Figure 2A). However, unlike the common *ent*-kaurane diterpenoids that contain 20 carbons in the skeletons, **1** was shown to have 23 carbons in its skeletal structure. The extra three carbons were found to belong to an isopropyl group, which formed a carbon-carbon bond directly connected to C-17, as determined by the analysis of the 1D and 2D NMR data. In addition, the COSY spectrum of **1** showed the chained cross-peak correlations between H_2_-1, H_2_-2, and H_2_-3, between H-9, H-11, H_2_-12, H-13, and H_2_-14, and between H-13, H-16, H_2_-17, H-21, H_2_-22, and H_2_-23, and the HMBC correlations from H-11 to C-8 and C-13, H-19 to C-3, C-5, C-6, C-18, H-20 to C-5, C-7 and C-18, H-21 to C-17, and C-22 and C-23. The stereochemistry of **1** was further elucidated by observing the spatially closed cross-peak signals of the protons in the NOESY spectrum. Because of the NOE correlations of H-11 with H-2*β*, H-9*β* and H-12*β*, and H-16 with H-13*α* and H-14*β*, the H-11 was determined to be *β*-oriented, whereas H-16 was *α*-oriented.

To confirm the absolute stereochemistry, **1** was dissolved in MeOH, followed by crystallization to afford a crystal suitable for an X-ray diffraction analysis. As a result of the final refinement, the Flack parameter of 0.00 (6) suggested its (4*R*, 5*R*, 8*S*, 9*S*, 10*R*, 11*R*, 13*S*, and 16*R*) absolute configuration (Figure 2B). Thus, compound **1** (guidongnin I) was defined as 16*β*-isopropyl-11*α*-hydroxy-6,7-*seco*-6,19:7,20-diolide-*ent*-kaur-15-one.

Compound **2** was isolated and finally obtained as colorless flaky crystals (MeOH, melting point: 223.9–226.1 °C), and the HRESIMS sodium adduct at *m*/*z* 369.1673 [M + Na]^+^ (calcd for C_20_H_26_O_5_Na, 369.1666) determined its molecular formula as C_20_H_26_O_5_. Its IR absorptions at *ν*_max_ 1722 and 3419 cm^−1^ implied the existence of carbonyl and hydroxyl groups in **2**. Based on the analysis of the ^13^C- and DEPT-NMR spectral data (Table 1), the 20 carbons in **2** were identified to be two olefinic carbons [*δ*_C_ 153.3 (C-16), 107.7 (C-17)], the carbonyl carbon of a lactone [*δ*_C_ 173.5 (C-7)], three non-oxygenated quaternary carbons [*δ*_C_ 40.2 (C-4), 51.1 (C-8), 44.5 (C-10)], three non-oxy methine carbons [*δ*_C_ 56.0 (C-5), 33.3 (C-9), 39.6 (C-13)], an oxy methine [*δ*_C_ 79.2 (C-15)], two acetal carbons [*δ*_C_ 106.7 (C-6), 104.1 (C-20)], six non-oxy methylene carbons [*δ*_C_ 26.4 (C-1), 17.1 (C-2), 30.6 (C-3), 18.0 (C-11), 32.7 (C-12), 37.9 (C-14)], one oxy methylene carbon [*δ*_C_ 76.3 (C-19)], and one methyl carbon [*δ*_C_ 31.0 (C-18)]. By searching for the literature compounds of *ent*-kaurenes, **2** was found to show similar ^1^H and ^13^C NMR data to the known compound macrocalyxoformin E [15]. The main structural difference between the two compounds was that **2** formed an extra tetrahydrofuran ring between C-6 and C-19, as evidenced by the presence of the HMBC correlations of H-6 to C-19 (*δ*_C_ 76.3) and of Ha-19 (*δ*_H_ 3.46, d, *J* = 9.0) to C-6 (Figure 3A). The presence of the NOE correlations of H-6 with H-5*β*, H-9*β* and H_3_-18, H-20 with H-2*α* and H-19*α*, and H-15 with H-14*β* suggested that H-6 was *β*-oriented, while H-15 and H-20 were *α*-oriented (Figure 3A). Furthermore, the X-ray diffraction analysis of **2** confirmed the relative configurations of **2** (Figure 3B) as 4*R**, 5*R**, 6*S**, 8*S**, 9*S**, 10*R**, 13*R**, 15*R**, and 20*R**. Compound **2** was thus identified as 15*β*-hydroxy-6,19: 6,20-diepoxy-6,7-*seco*-7,20-olide-*ent*-kaur-16-en, and given the trivial name guidongnin J.

Plausible biogenetic pathways for compounds **1** and **2** are postulated in Figure 1. Guidongnins I (**1**) and J (**2**) could be formed through a series of reactions starting from kau-16-en-15-one (**A**), the normal *ent*-kaurene that commonly exists in *Isodon* plants. The oxidations at C-7, -19, and -20 of A produce compound **B**, which undergoes dehydration to form a double bond between C-6 and C-7 by losing an H_2_O molecule. The cleavage of the 6,7 double bond in **C** by dioxygenase leads to the formation of compound **D** with two carbaldehyde groups. **D** then undergoes two separate biogenetic pathways to produce **1** and **2**. When the carbaldehyde at C-5 in **D** reacts with 19-OH to form the first tetrahydrofuran group through a hemiacetal linkage, followed by the 2nd and 3rd hemiacetal formations between C-6 and C-20 (**J**) and between C-20 and C-7 (**K**), respectively, compound **L** with a three-fused cyclic ether ring system is yielded. Further oxidation of the C-7 acetal group, followed by the reduction of the carbonyl carbon at C-15, **L** is transformed to compound **2** through **M**. On the other hand, when hemiacetal conversions take place for both carbaldehyde groups in **D** at the same time, compound **E** is produced. With the subsequent oxidations at C-7 and C-11, **E** is converted to ludongnin A (**G**), a major component (0.63%) in Isodon rubescens Hara var lushiensis [16]. Ludongnin A (**G**) further reacts with isobutyryl-CoA via a Michael-like addition reaction to afford **H**. Upon losing CO_2_, compound **1** is eventually produced from **H**.

### 2.2. Biological Activity

It is reported that the *I. rubescens* extract was found to inhibit a variety of cancer cells [17]. Previously, the diterpenoids isolated from this species, such as oridonin, rosthorin, lushanrubescensin H, and rabdosin A, showed significant inhibitory activity against the growth of tumor cell lines [18]. Therefore, guidongnins I (**1**) and J (**2**) were evaluated for their cytotoxicities against various human cancer cell lines [A-549 (lung), HL-60 (leukemia), MCF-7 (breast), HepG2 (liver), and COLO-205 (colon)] using the CCK-8 method as previously reported [19,20,21]. The two compounds were inactive at a concentration of 100 µM, except for **2** on HepG2 cells. Compound **2** was found to show antiproliferative activity against HepG2 cells at an IC_50_ of 27.14 ± 3.43 µM. Since the carbonyl group is reduced to a hydroxy group and **2** contains no *α*, *β*-conjugate system, which is normally required for the biological activity of an *ent*-kaurane diterpenoid, the antiproliferative activity of **2** against HepG2 cells could arise from the three-fused ring system. Further study of the fused ring system with an activity relationship is thus needed to explore this unique structural unit of *ent*-kaurene compounds.

## 3. Materials and Methods

### 3.1. General Experimental Procedures

An XRC-1 micro-melting point apparatus (Sichuan Weitai, Chengdu, China) was used for the measurement of the compounds’ melting points. Optical rotations were determined using a Jasco P-1010 polarimeter (Perkin-Elmer, Waltham, MA, USA). The IR spectra were recorded on a VECTOR22 infrared spectrometer using the KBr pellets method. A Bruker Avance spectrometer was used for recording the NMR spectra. The HR-ESI-MS data were obtained using a Bruker Q-TOF mass spectrometer (Bruker, Karlsruhe, Germany). X-ray data were collected on a Bruker APEX-II CCD instrument (Bruker, Rheinstetten, Germany) equipped with Cu *Kα* radiation for **1** and a Bruker APEX-II instrument equipped with Mo *Kα* radiation for **2**. Silica gel (200–300 mesh) was used for column chromatography, while silica gel GF_254_ (0.2 mm) was used for analytical TLC. Prior to use, all solvents were distilled. Detailed information is displayed in the Appendix A.

### 3.2. Plant Material

*I. rubesens* twigs and leaves were collected from Guiyang, Guizhou Province, China, in July 2018. The plant material was identified by Professor Junhua Zhao, and a voucher specimen (No. 20180713) was deposited at Guizhou University of Traditional Chinese Medicine.

### 3.3. Extraction and Isolation

*I. rubescens* twigs and leaves (8.8 kg) were dried, powdered, and extracted with MeOH (3 × 10 L) at room temperature to give a crude extract (1.3 kg) after concentration. The resultant residue was suspended in H_2_O, which was then extracted with petroleum ether (1 L × 2) and EtOAc (1 L × 3). Subsequently, the EtOAc-soluble fraction (230.2 g) was divided into 6 fractions (A-F) using silica gel column chromatography, eluting with gradients of petroleum ether/EtOAc (100:1–0:1). Fraction C (37.4 g) was applied on a RP-8 (octylsilane bonded silica gel) column (MeOH/H_2_O, 5:95–50:50) to produce subfractions C1-C4. Fraction C2 (6.8 g) was separated by silica gel column chromatography (CH_2_Cl_2_/EtOAc, 100:1–1:1) to obtain compounds **1** (20 mg) and **2** (14 mg), followed by recrystallization in MeOH to yield suitable crystals for X-ray crystallographic analysis.

Guidongnin I (**1**): colorless flaky crystals; m.p. 247~248 °C; ${\left[\alpha \right]}_{D}^{21.3}$ −127.27 (c 0.2200, CH_2_Cl_2_); IR (KBr) *ν*_max_ 3380, 2941, 2864, 2357, 1749, 1697, 1468, 1106, 1059, 1033, 868 cm^−1^; 1D NMR data, see Table 1; HR-ESI-MS *m*/*z* 427.2091 [M + Na]^+^ (calcd. for C_23_H_32_O_6_Na, 427.2086).

Guidongnin J (**2**): colorless transparent sheet-shaped crystals; m.p. 223.9~226.1 °C; ${\left[\alpha \right]}_{D}^{21.3}$−32.92 (c 2.4300, MeOH); IR (KBr) *ν*_max_ 3419, 2938, 1723, 1458, 1417, 1297, 1252, 1125, 1086, 1057, 980 cm^−1^; 1D NMR data, see Table 1; HR-ESI-MS *m*/*z* 369.1673 [M + Na]^+^ (calcd for C_20_H_26_O_5_ Na, 369.1666).

### 3.4. X-ray Data of Compounds ***1***–***2***

Colorless block crystals of guidongnins I–J (**1**–**2**) were obtained from the MeOH solution. Crystallographic data for **1** were collected on a Bruker APEX-II CCD instrument using Cu *Kα* radiation, and for **2** were collected on a Bruker APEX II CCD instrument using Mo *Kα* radiation (Table 2). With the use of Bruker SAINT, cell refinement and data reduction were accomplished, where the structure was solved by means of SHELXTL [22,23]. The crystallographic data for **1** (CCDC 2288451), **2**, and (CCDC 2261016) can be obtained on request at www.ccdc.cam.ac.uk/data_request/cif (accessed on 10 May 2023).

### 3.5. Cytotoxicity Assay of Compounds ***1***–***2***

The anti-proliferative activities of compounds **1**–**2** against human cancer cell lines were evaluated using the CCK-8 method, as previously described [18,19]. In the CCK-8 assay, the A549 cells were grown in Roswell Park Memorial Institute (RPMI) 1640 medium. HL-60, MCF-7, HepG2, and COLO-205 cells were grown in Dulbecco’s modified Eagle’s medium (DMEM). All used medium was supplemented with 10% fetal bovine serum (FBS) in a 37 °C incubator with a humidified atmosphere containing 5% CO_2_. A density of 2 × 10^5^ cell/mL cancer cells (200 μL) were seeded in 96-well microtiter plates and exposed to 200 μL of various concentrations of compounds in triplicate for 72 h. The initial concentrations of the compounds were 10.0 mM in DMSO and then diluted into 20, 10, 5, 2.5, 1.25, and 0.625 μM in RPMI-1640/DMEM medium. Then, 100 μL of the test solutions were added to each well and further incubated for 72 h. The CCK-8 solution (10 μL) was then added to each well and incubated for 2–3 h. Absorbance was then measured at 450 nm.

## 4. Conclusions

In conclusion, the present phytochemical study of *I. rubescens* resulted in the isolation of two novel 6,7-*seco*-7,20-olide-*ent*-kaurene diterpenes (**1** and **2**). Their chemical structures were elucidated by means of comprehensive spectroscopic data analysis (Appendix A) and further confirmed by single-crystal X-ray crystallographic data. The isolates were evaluated for their bioactivities on a panel of human cancer cell lines. Compound **2** demonstrated antiproliferative activity against HepG2 cells, which may warrant **2** as a lead molecule for further chemical and biological exploration since it contains a three-fused ring system instead of *α*,*β*-conjugate system that is normally required for the biological activity of an *ent*-kaurane diterpenoid.

## Data Availability

The data presented in this study are available on request from the corresponding author.

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
