# Peer review of "Guidongnins I–J: Two New 6,7-seco-7,20-Olide-ent-kaurene Diterpenes with Unusual Structures from Isodon rubescens"

_ijms, 2023, doi:10.3390/ijms241713451_

Round 1
Reviewer 1 Report
The revised manuscript presents spectroscopic and single-crystal X-ray studies of two novel 6,7-seco-7,20-olide-ent-kaurene diterpenes, Guidongnin I (1) and Guidongnin J (2), isolated from the plant Isodon rubescens.
The special issue of Molecular Pharmacology IJMS entitled Natural Bioactive Compounds for Human Health 2.0 is the right choice considering
cytotoxic aspects (antiproliferative activity of compound 2 against HepG2 cells).
The research topic examined in this manuscript looks quite interesting and the present paper is worth publishing but it needs major revision due
to lack of diligence.
Comments:
1) The Authors describe the structure of Guidongnin K although they only isolated Guidongnins I and J.
2) According to CIF files obtained on request from the Cambridge Crystallographic Data Centre the formula and molecular weight
of the first compound are wrong.
3) The wavelength (1.34138 A) used in the refinement process of 1 does not correspond to copper radiation.
4) If Ga K (alpha) was used for 1 it should be corrected in the manuscript.
5) Molybdenum radiation is not recommended for determining the absolute structure of a compound that has only such atoms as carbon, hydrogen and oxygen (see compound 2).
A minor editing of the English language is required.
Reviewer 2 Report
The manuscript reported the isolation of two diterpenes isolated from the medicinal plant Isodon rubescens. Their structures and absolute configurations were determined by the analyses of the spectroscopic and X-ray diffraction data. Guidongnins I (1) and J (2) were evaluated for their cancer cell killing activities against a panel of cancer cell lines, and compound 2 showed a low cytotoxic profile against HepG2 with an IC50 value of 27.14±3.43 μM. The manuscript deserves to be published only for the identification and characterization of two new ent-kaurane diterpenes from Isodon rubescens. I have two main criticisms.
a) Several ent-kaurane diterpenes, among other compounds, were previously isolated were isolated from the leaves of Isodon rubescens (J. Nat. Prod. 2003, 66, 1391-1394) and all compounds characterized by the presence of cyclopentanone conjugated with an exomethylene group as Michael acceptor was the active center of the inhibitory effect against K562 cells. The two reported compounds were characterized by the absence of this moiety and this explain the low antiproliferative activity of the tested compounds.
b) In the extraction and isolation section, can the authors explain with more details if the authors identified only two compounds by column chromatography on fraction C? Fractions A, B and D-f were investigated?
Reviewer 3 Report
The manuscript “Guidongnins I-J, two new 6,7-seco-7,20-olide-ent-kaurene diterpenes with unusual structures from Isodon rubescens“ [ijms-2551486] written by Juan Zou, Jiang-Hai Ye, Chen-Liang Zhao, Jing-Jie Zhang, Ya-Hua Liu, Lu-Tai Pan, Kang He and Hong-Jie Zhang describes the isolation and structural identification of two new diterpenoids from leaves of the plant Isodon rubescens. In particular 1D and 2D NMR, MS and x-ray analysis have been used to determine the structures. Possible biosynthesis of the two compounds is discussed. Based on anti cancer activities of crude extracts from Isodon rubescens, both compounds have been tested for their cytostatic activities against a couple of cell lines.
The reviewer has expertise in the molecular field of organic chemistry and structure determination (in particular NMR) and hence mostly refers to this part of the manuscript with the review.
All structural investigations and biological test seem to be performed with modern and common state of the art methods. The overall practical work seems quite well planned and performed. The practical investigation has been made carefully with state of the art methods. The further analysis of the data as well as the discussion and conclusion are sensible in many parts and detailed with respect to data gained. Some concerns about the presentation of the proposed biosynthesis are given in the comments.
Hence, the results possess some importance in furthering our knowledge of diterpenoids from Lamiaceae, in particular from Isodon rubescens. The manuscript is therefor of some interest in the fields of Phytochemistry, Organic Chemistry, Natural Product Chemistry and to some extent of Medicinal Chemistry and Pharmacy. The results are worth publishing in general. However, there are some points listed below, which should be further addressed by the authors. Hence, there are some comments listed below, which should be taken into account by the authors prior to acceptance of the manuscript. Therefore, the manuscript is not yet in a form to be published in “International Journal of Molecular Science“.
Major Comments:
1) In "Results and Discussion" the authors start directly with the description of the structure. Here it would be appropriate if a short paragraph on isolation is also inserted in order to present this part of the work as a result. Care should be taken to show that the two compounds are not isolation artifacts.
2) The presentation of the possible biosynthesis is incomprehensible. The authors describe possible intermediates with the letters A-H without making these structures clear in an according graphic. In this associated graphic, on the other hand, a structure with the number "3" appears, which is not mentioned in the text. The authors are therefore strongly encouraged to thoroughly revise this part of the manuscript and to present the possible biosynthesis in such a way that it is comprehensible for the reader. The following should be observed:
2a) At least one reference should be given for the postulated Michael reaction, which documents such a reaction in a biosynthesis. This is necessary in order not to postulate fundamentally undescribed steps of a possible biosynthesis.
2b) Is there a described (or postulated) biosynthesis of "compound 3" and is it certain that the two new compounds described are very likely to originate from this "compound 3"? If this is not the case, the sequence "reduction, oxidation, ketalization" should be reconsidered. Based on the data from the manuscript, the reviewer would assume macrocalyxoformin C to be a possible "starting material" for this part of the biosynthesis.
3) Authors are encouraged, in Supp. Mat. to present the (original) data of the X-ray structure analysis.
4) The cytostatic activity does not appear to be extremely pronounced to the reviewer. The authors are therefore encouraged to address this more intensively in the discussion and to discuss it in the context of the anti-cancer activities of crude extracts from Isodon rubens. Possibly it would be possible to check the two compounds for other bioactivities and thus to be able to make statements about the possible importance of the compounds for the plant or about possible other pharmaceutical uses. (In the case of the latter, it should be noted that the amount recovered is very small and thus access seems difficult.)
Minor comments:
5) Line 198 "H2O" with subscript "2".
6) The numbers of the two compounds 1 and 2 should always be printed in bold.
7) The phrase "cancer cell killing" in the abstract does not sound very scientific and its use should be reconsidered.
Round 2
Reviewer 1 Report
Due to the change in the CCDC number for compound 1, I kindly ask the Authors to enclose the CIF and CHECKCIF files.
In paragraph 3.4 instead of "flack parameter" should be "Flack parameter".
Reviewer 2 Report
The manuscript deserves to be published in the present form.
Reviewer 3 Report
The manuscript “Guidongnins I-J, two new 6,7-seco-7,20-olide-ent-kaurene diterpenes with unusual structures from Isodon rubescens“ [ijms-2551486-v2] written by Juan Zou, Jiang-Hai Ye, Chen-Liang Zhao, Jing-Jie Zhang, Ya-Hua Liu, Lu-Tai Pan, Kang He and Hong-Jie Zhang has been revised and resubmitted by the authors. The reviewer is grateful to the authors for taking to heart the previous comments, also from other reviewers and for addressing several concerns.
A list with answers of the authors to the reviewer’s comments, point by point, is added to the answer. The authors have taken all of the comments of reviewer 3 and obviously of other reviewers into account and made the respective corrections, additions, and changes in the revised manuscript.
The equality and readability of the manuscript have hence been improved in the newly submitted version. The manuscript hence now fully and intelligibly describes two new diterpenes from Isodon rubescens including structure determination and possible biosynthesis as well as some anti-cancer activities of these compounds. The manuscript is therefore acceptable for publication in the “International Journal of Molecular Science”.
Round 3
Reviewer 1 Report
Subsection 3.4 has been substantially revised and Table 1 has been added.
Comments;
1) Z' values ​​should be given in Table 1 for both compounds.
2) Could you correct the temperature given in Table 1 for crystal 1?
3) Compound 2 crystallizes in space group P2(1) with two molecules in the asymmetric part of the unit cell. Only one selected molecule is shown in Figure 3 and any explanation is given. Do they differ from each other?
For verification, could you please, enclose a figure with superimposed molecules?
Round 4
Reviewer 1 Report
Thank you very much for the explanation regarding the reason for choosing only one molecule for the graphical presentation in the case of compound 2. A minor note: The attached figure is not the effect of superimposing two molecules of compound 2.
In Table 2 I expected a row with the following notation: Z, Z' 4, 1 4, 2 to show that in structure 2 there are two molecules in the asymmetric part of the unit cell. If the Authors want to stay with the previous entry, it is their choice.
I also noticed the removal of the Flack parameter from Table 2.